# EGSST: Event-based Graph Spatiotemporal Sensitive Transformer for Object Detection

**Sheng Wu[1]**    **Hang Sheng[1]**    **Hui Feng[1,2]** *    **Bo Hu[1,2]**

[1] School of Information Science and Technology, Fudan University
[2] State Key Laboratory of Integrated Chips and Systems, Fudan University

## Abstract

Event cameras provide exceptionally high temporal resolution in dynamic vision systems due to their unique event-driven mechanism. However, the sparse and asynchronous nature of event data makes frame-based visual processing methods inappropriate. This study proposes a novel framework, Event-based Graph Spatiotemporal Sensitive Transformer (EGSST), for the exploitation of spatial and temporal properties of event data. Firstly, a well-designed graph structure is employed to model event data, which not only preserves the original temporal data but also captures spatial details. Furthermore, inspired by the phenomenon that human eyes pay more attention to objects that produce significant dynamic changes, we design a Spatiotemporal Sensitivity Module (SSM) and an adaptive Temporal Activation Controller (TAC). Through these two modules, our framework can mimic the response of the human eyes in dynamic environments by selectively activating the temporal attention mechanism based on the relative dynamics of event data, thereby effectively conserving computational resources. In addition, the integration of a lightweight, multi-scale Linear Vision Transformer (LViT) markedly enhances processing efficiency. Our research proposes a fully event-driven approach, effectively exploiting the temporal precision of event data and optimising the allocation of computational resources by intelligently distinguishing the dynamics within the event data. The framework provides a lightweight, fast, accurate, and fully event-based solution for object detection tasks in complex dynamic environments, demonstrating significant practicality and potential for application. The source code can be found at: EGSST.

## 1 Introduction

There is an increasing demand for devices capable of accurately capturing targets in high-speed dynamic scenes, such as autonomous driving, where traditional CMOS or CCD visual sensors often encounter motion blur and overexposure [1, 2, 3]. In response, event cameras, which employ a novel event-driven mechanism, have garnered significant attention. Each pixel in event cameras operates independently, activating only upon detecting a brightness change. This mechanism results in a series of high-speed, asynchronous event streams with exceptionally high temporal resolution, which enables the rapid and precise capture of data [4, 5].

Our objective in processing the output data from event cameras is to effectively utilize their high temporal precision in detection tasks without inducing significant delays. However, the data format of event cameras is entirely different from that of frame-based cameras, rendering existing object detection methods based on RGB images inapplicable [6]. Some methods adopt conversion strategies, utilizing techniques such as temporal slicing to transform sparse data into dense formats, with

---

*Corresponding author.

38th Conference on Neural Information Processing Systems (NeurIPS 2024).

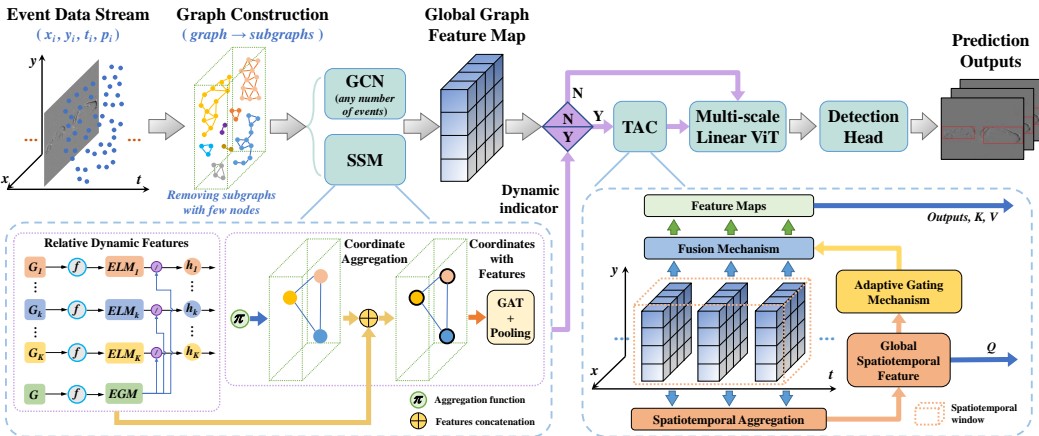

Figure 1: **An overview of the proposed EGSST framework.** The EGSST is an event-based, lightweight, and efficient framework designed for rapid object detection in event data. A graph is constructed from the data and divided into $K$ connected subgraphs. These subgraphs are fed into a Graph Convolutional Network (GCN) [22] and a SSM. The GCN processes the subgraphs that are not removed to produce a global Graph Feature Map, which preserves both spatial and temporal information. The SSM assesses the dynamics of the entire graph and outputs a dynamic feature indicator, which includes the dynamics of each subgraph and the aggregated dynamics obtained through a Graph Attention Network (GAT) [23]. The TAC is activated based on the output from the SSM to enhance focus on the temporal dimension or to feed the graph feature maps directly into a Multi-scale Linear ViT [24]. Finally, a detection head, such as RT-DETR [25] or YOLOX [26], is employed to generate prediction outputs.

the aim of leveraging architectures like Convolutional Neural Networks (CNNs) or Transformers [2, 7, 8, 9, 10, 11]. Unfortunately, methods employing such conversion strategies often result in a loss of temporal precision. Another direct processing strategies involve methods such as Graph Neural Networks (GNNs) [12, 13, 14, 15] or Spiking Neural Networks (SNNs) [16, 17, 18, 19, 20, 21], which analyze the event stream directly, thereby preserving spatiotemporal structure and accuracy. However, the direct processing strategies often suffer from computational efficiency. The latest hybrid strategies strive to combine the strengths of previous methods, enhancing performance while maximizing retention of temporal precision. Although numerous existing strategies have made significant progress in the extraction of event data features, challenges remain in effectively managing data across spatial and temporal dimensions [12, 14].

Existing methods may result in the inefficient use of computational resources when processing data across both spatial and temporal dimensions. Because current models either apply a single algorithm to both dimensions simultaneously [12, 20] or use different algorithms for each dimension and then combine the results [27]. Importantly, processing in the temporal dimension typically consumes significantly more computational resources than processing in the spatial dimension. Consequently, the continuous processing of temporal data is likely to result in the inefficient utilization of computational resources, especially for slower-moving objects where spatial resolution may already be sufficient.

In order to address the aforementioned issue of inefficiency, we have drawn inspiration from the dynamic perception of human eyes. The human visual system naturally prioritizes rapidly moving objects within the visual field while de-prioritizing slower ones—an adaptive feature that enhances responsiveness and efficiency in dynamic environments. Ideally, artificial models can emulate this trait by preferentially processing faster-moving objects in the temporal dimension, especially when managing large volumes of continuous spatiotemporal data. Therefore, we try to develop a novel algorithm that is closer to the selective attention mechanism of the human eye, which could potentially enhance both the efficiency and effectiveness of event-based vision systems.

We propose a novel spatiotemporal fusion graph transformer framework, designed to fully leverage the powerful capability of graph to process unstructured data while also reducing unnecessary computations in handling spatio-temporal data. Firstly, the framework employs a graph structure to

model event data, thereby maintaining the original temporal fidelity and capturing crucial spatial details. A key aspect of our framework is the SSM module, which leverages the graph's ability to process unstructured data and aggregates features to effectively discern the relative dynamics of objects. This differentiation provides a critical basis for determining whether to continue processing in the temporal dimension. Subsequently, the adaptive TAC is introduced, which dynamically adjusts its activation based on the insights from the SSM. The TAC is designed to enhance the processing of highly dynamic events while reducing resource use in scenarios with low dynamics. Furthermore, the integration of a lightweight multi-scale LViT markedly enhances the processing efficiency of our system. The principal contributions of our research are as follows:

- Our model introduces a novel and efficient graph processing method by pioneering the use of connected subgraphs in the context of event cameras. This technique not only preserves the temporal data completely but also enhances the effective and precise focus on targets within the event data.

- We introduce SSM and TAC to mimic the human eye's perception of dynamics and integrate them into the Graph Transformer framework for efficient object detection in event data.

- Our model integrates Graph and Transformer technologies to enhance object detection tasks in a fully event-based manner. It is designed to be lightweight, fast, and precise, representing a novel approach that leverages the strengths of both technologies for improved performance.

## 2 Related Work

The unique event-driven mechanism of event cameras offers considerable potential for high-speed motion detection in dynamic environments. However, the inherent sparsity and unstructured nature of the data generated by these cameras pose significant challenges to conventional image processing techniques [4, 28]. In response, researchers have transitioned from a singular processing strategy to a hybrid approach, integrating traditional and innovative methods.

**Conversion Processing Strategy** involves transforming event data into dense frame-based data, making it compatible with conventional visual processing algorithms. Systems like the E2VID [29] utilize CNNs to convert asynchronous event data into video frames through time slicing. Other approaches generate continuous optical flow and intensity estimation images from event streams [30], or incorporate attention mechanisms with recurrent and convolutional networks to enhance spatio-temporal feature extraction [31]. These strategies extend the applicability of event camera data but often at the cost of reducing its high temporal resolution [32, 33, 34].

**Direct Processing Strategy** focuses on preserving the original asynchronous characteristics of event data to maintain its sparsity and high temporal precision. This includes direct modeling of the event stream for real-time object tracking [35] and employing spiking neural networks (SNNs) for gesture recognition [7]. Additionally, models like AEGNN [12] leverage GNNs and efficient updating strategies to retain asynchronous temporal features. While these methods effectively utilize the unique properties of event data, they face challenges in handling large volumes of sparse data, highlighting the need for further innovations in architecture [36, 20].

**Hybrid Processing Strategy** explores a combination of conversion and direct processing methods, or the integration of multiple model frameworks [37, 38, 39, 40, 9, 27]. For example, merging GNNs with CNNs, and integrating Transformers with recurrent neural networks, effectively captures spatio-temporal features from asynchronous events while maintaining high resolution and enhancing adaptability [38, 39]. Methods like MatrixLSTM [40] and RED [9] increase data processing efficiency by jointly extracting features. Building on these studies, a recent work, RVT [27], integrates Transformers with recurrent neural networks, significantly improving processing efficiency and accuracy. However, current models, despite their commendable performance and efficiency, cannot effectively differentiate dynamic data, leading to unnecessary computations in the time dimension.

Our research lies within the hybrid processing strategy methods. The proposed framework leverages graph capabilities to manage irregular data and incorporates a temporal attention mechanism responsive to data dynamics. It also integrates a lightweight linear visual Transformer to extract temporal and spatial information from event data. This event-based framework is designed to provide a fast, efficient, and accurate object detection solution, specifically tailored to event cameras.

# 3 Methodology

The proposed framework utilizes the Graph Transformer to extract graph-based features from unstructured event data and integrates both spatial and temporal attention mechanisms. As illustrated in Figure 1, the architecture and key components are depicted, providing a visual overview of how these elements are integrated. This section will outline the critical steps involved in the framework.

## 3.1 Graph Construction

Each event captured by an event camera records both spatial information and precise temporal details, crucial for dynamic scene analysis. That is, each event is recorded as

$$event_i = (x_i, y_i, t_i, p_i), \quad i \in \mathcal{N}, \tag{1}$$

where $x_i, y_i$ locate the pixel and $t_i$ represents the timestamp with microsecond precision. The coordinates $x_i$ and $y_i$ as well as the timestamps $t_i$ are completely asynchronous and occur randomly, reflecting the event-driven nature of the data capture. The polarity $p_i \in \{-1, 1\}$ denotes whether there is a decrease (for $-1$) or an increase (for 1) in pixel brightness. $\mathcal{N} = \{1, 2, \ldots, N\}$ is the set of indexes of events.

A graph-based representation is employed to capture irregular spatiotemporal relationships between events. Firstly, in order to enhance the stability of the model and prevent data bias, especially for timestamps with otherwise large values, it is necessary to normalize the timestamp for each event by the equation $t_i^* = \beta \cdot (t_i - t_0), i \in \mathcal{N}$, where $t_0 = \min_{i \in \mathcal{N}} \{t_i\}$ and $\beta$ is a normalization factor. Each event $event_i$ then generates a vertex $v_i = (x_i, y_i, t_i^*, p_i)$, and $V = \{v_1, v_2, \ldots, v_N\}$ is the set of vertices. The position coordinates of each vertex can be represented as $\mathbf{c}_i = [x_i, y_i, t_i^*]^T$.

To establish the graph edges, we consider every pair $(\mathbf{c}_i, \mathbf{c}_j)$. An edge $e_{ij}$ is added to the set $E$ of edges if the conditions are met,

$$e_{ij} = \begin{cases} 1, & \text{if } \|\mathbf{c}_i - \mathbf{c}_j\| \le R, \\ 0, & \text{otherwise}, \end{cases} \tag{2}$$

indicating the connectivity based on the predefined spatiotemporal distance threshold $R$.

The resulting graph, $G = (V, E)$, effectively encapsulates the essential spatialtemporal properties of events.

## 3.2 Connected Subgraphs Construction

Event cameras produce data triggered by pixel brightness changes. Dense events occur at object contours where brightness exhibits a significant variation, whereas the main body sees fewer, sparse events due to subtle changes. Transforming this event data into graph $G$, areas with dense events display numerous edges, whereas sparser areas show fewer edges. This pattern of data density leads us to the concept of connected graphs in graph theory, which describes the interconnectivity between the nodes of an undirected graph. The contour regions, due to their high number of edges, form several large connected subgraphs with a substantial number of nodes. In contrast, the main bodies, which have fewer edges, result in connected subgraphs with fewer nodes. By filtering the data based on the number of nodes, the majority of the retained connected subgraphs attain new physical meaning, indicating the contiguous edges of object contours. The entire graph $G$ can be divided into $K$ connected subgraphs, $G_k = (V_k, E_k)$, which satisfy

$$\mathcal{D} = \{G_k \colon G_k \subseteq G, \quad k = 1, 2, \cdots, K\}. \tag{3}$$

However, it should be noted that the connected subgraphs are not in one-to-one correspondence with the objects $D_l$ in the event data. Consequently, the number of selected subgraphs, $K$, is typically greater than the number of real objects, $L$.

$$\mathcal{D}_l \subseteq \mathcal{D}, \quad l = 1, 2, \cdots, L \tag{4}$$

By filtering the number of nodes contained in the connected subgraphs, we can filter out subgraphs containing too few nodes, thus realizing downsampling. Based on preliminary experimental tests with the dataset used, processing 10,000 events can retain approximately 73% of the events, which

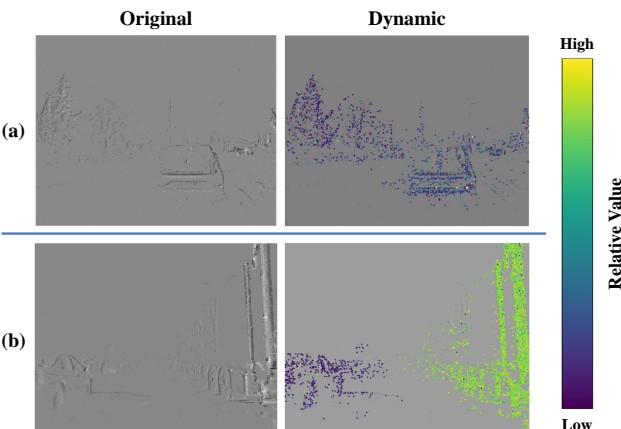

Figure 2: **Dynamic Visualization of the SSM.** Each image is generated from 10,000 event points, causing slight blurring. However, connected subgraphs effectively filter out background noise, preserving only relevant objects. (a) The scene shows low relative dynamic, hence the distinction is not pronounced. (b) The truck on the right accelerates to overtake, while the car on the left moves slower, making the truck's relative values significantly higher.

helps reduce noise interference. In contrast to conventional uniform downsampling, this approach avoids the loss of valid features associated with nodes. Moreover, the scope of event data that requires focus can be selected in a more effective and rational manner.

After constructing connected subgraphs, we utilize a multilayer GCN that integrates node feature information with graph topology, efficiently learning node representations. The GCN preserves and fuses spatial and temporal features globally, producing a global graph feature map.

### 3.3 Spatiotemporal Sensitivity Module (SSM)

The Spatiotemporal Sensitivity Module (SSM) is a core component of our framework. It mimics the visual characteristics of the human eye to quantify object dynamics in event data. Since event data differs from traditional images, directly quantifying object dynamics using displacement speed is challenging. Thus, we propose Event Global Motion (EGM) and Event Local Motion (ELM), applied to the global graph $G$ and subgraphs $G_k$, respectively.

We first define the following metrics to separately quantify the global and local dynamics:

For global motion, we define:
$$EGM = f(N, \Delta t^*), \tag{5}$$

where $N = |\mathcal{N}|$ represents the number of nodes in the global graph, and $\Delta t^* = \max_{i,j \in \mathcal{N}}\{|t_i^* - t_j^*|\}$ represents the maximum time difference among nodes.

For local motion, we define:
$$ELM_k = f(n_k, \delta t_k^*), \tag{6}$$

where $n_k$ represents the number of nodes involved in the measurement of the $k$-th subgraph, and $\delta t_k^*$ is the maximum time difference within the subgraph. While the function $f$ could be designed to take more complex forms to capture intricate relationships between these variables, we opt for a simple ratio-based method, $f(x, y) = x/y$. This choice is motivated by the desire to balance simplicity and computational efficiency. The ratio directly reflects the relationship between the number of events and the temporal spread in the subgraph, providing an intuitive measure of local motion dynamics. By using this straightforward formulation, we ensure that the computation remains tractable, especially when dealing with large-scale event data, while still effectively capturing the key dynamics of interest.

For instance, in a traffic scenario, event data may include dynamic entities such as vehicles and pedestrians, as well as static background elements like trees and fences. $EGM$ quantifies the dynamics of the entire event data, representing the overall level of motion in the environment, while $ELM_k$ describes the motion level of individual subgraphs. Given the high temporal resolution of

event cameras, we can assume that any object responds to events almost instantaneously. As an object's speed increases, more pixels detect brightness changes, generating more events, which increases $n_k$ and decreases $\delta t_k^*$, leading to a significant increase in $ELM_k$.

To analyze the relationship between the dynamics of each subgraph relative to the overall environment, we introduce the following metric:

$$H = \left\{ h_k = \frac{ELM_k}{EGM} : k = 1, 2, \ldots, K \right\}, \tag{7}$$

which quantifies the relative motion of each subgraph with respect to the global dynamic level. During subgraph construction, distance thresholds and limitations on adjacent nodes are set to divide large objects into multiple subgraphs, thus preventing a single subgraph from dominating the motion representation of a large object.

While all nodes in each subgraph should reflect relative dynamics, assigning the same features to every node is unnecessary as it would result in redundant computations. We aggregate the spatiotemporal coordinates of each subgraph into representative coordinates as follows:

$$S = \left\{ s_k = \pi(V_k^*) : k = 1, 2, \ldots, K \right\}, \tag{8}$$

where $V_k^*$ denotes the set of spatiotemporal position coordinates for subgraph nodes, and $\pi$ is a function for aggregation [12, 41], such as mean, max, or min. These functions are chosen because they are insensitive to the varying number of nodes in each subgraph, ensuring that the aggregation process remains consistent regardless of subgraph size. We then use the $K$-nearest neighbors algorithm to connect these new representative coordinates, forming a new set of edges $\mathcal{E}$, mapping features $h_k$ to the corresponding representative coordinates, and building a new set of vertices $\mathcal{V}$. This results in a new graph $\mathcal{G} = (\mathcal{V}, \mathcal{E})$, which significantly reduces the number of nodes and decreases computational complexity and processing delay.

In the newly constructed graph, each node represents an aggregated subgraph with a relatively sparse distribution in the spatiotemporal domain, necessitating the enhancement or suppression of relationships between nodes. We introduce inter-subgraph attention using a Graph Attention Network (GAT) [23] to capture relationships among nodes. The GAT enables nodes to adaptively aggregate information based on the importance of their features and their neighbors, strengthening the connections between nodes representing the same object while weakening those representing different objects. The GAT's attention mechanism also ensures that smaller objects are not overlooked and evaluates their significance in the overall environment. The aggregated values are used as the final output of the SSM for assessing relative motion. Figure 2 shows some example results, illustrating how the spatiotemporal sensitivity module captures the relative dynamics in different regions.

### 3.4 Temporal Activation Controller

Upon evaluation of the output from the SSM, a determination is made as to whether to enhance the temporal focus by activating the TAC. The TAC processing comprises two parts: firstly, a simple attention-like method is employed to dynamically weight and selectively emphasise the temporal dimension; secondly, the Query, Key, and Value (QKV) are reconstructed.

In the first part of the TAC, we define a set $\mathcal{F} = \{F_t\}$, where $F_t$ represents the stream of graph feature maps over time $t$, and $\mathcal{F}$ is the collection of these streams. We aggregate all features within the set using the formula $F_{st} = Aggregation\{\mathcal{F}\}$ to generate the Global Spatiotemporal Feature, $F_{st}$, which serves as a comprehensive representation of the spatiotemporal data. Subsequently, we employ an Adaptive Gating Mechanism to adjust these features at each timestamp, generating gating signals from $F_{st}$ for each $F_t$. This mechanism either enhances or suppresses the features based on the global feature, thereby maintaining temporal integrity while adapting to the sequence context. Finally, an adaptive fusion technique modulates the integration between the original and enhanced features, producing outputs suitable for subsequent processing.

In the second part of TAC, different sources generate the QKV matrices,

$$\begin{aligned} Q &= W_q(F_{st}), \\ K, V &= W_{k,v}(\mathcal{F}). \end{aligned} \tag{9}$$

The global spatiotemporal feature $F_{st}$ is employed to generate the Query (Q), which offers a global perspective when assessing the relevance of each timestamp to the entire sequence. Meanwhile, the

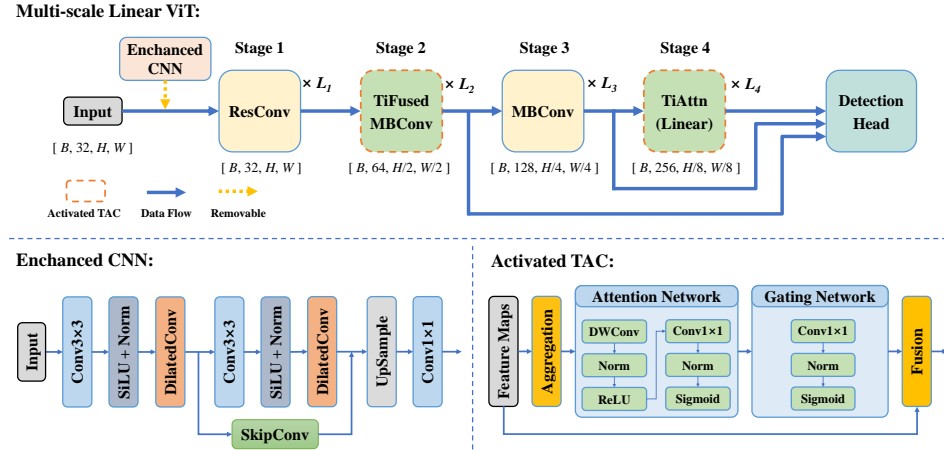

Figure 3: **The flowchart of the Multi-scale Linear ViT.** This diagram shows the stages of the Multi-scale Linear ViT, with the removable Enhanced CNN and Activated TAC modules. The Enhanced CNN processes input data through convolutional and normalization layers before passing it to the ViT stages. Activated TACs at Stage 2 and Stage 4 optimize temporal processing and balance efficiency. The data is then sent to the detection head for final object detection.

utilisation of the aggregated outputs of each timestamp in the first part as the source of Keys (K) and Values (V) generation helps to reflect the state of each timestamp with greater accuracy. This method of employing diverse generation sources merges global and local insights, thereby bolstering the model's capability to process complex spatiotemporal data within a self-attention architecture.

It is worth mentioning that TAC is not a module but a component, so it can theoretically be loaded on any model focusing on the extraction of spatial features, greatly improving scalability.

### 3.5 Multi-scale Linear ViT

The original Vision Transformer (ViT) [42] does not allow for multi-scale processing, resulting in significant performance degradation compared to models equipped with such capabilities [43, 44, 45]. To address this issue, our enhanced ViT module supports multi-scale processing and incorporates the TAC, which improves the model's ability to handle spatial and temporal dimensions. To reduce the computational load and increase processing speed, we adopt linear transformations to simplify the self-attention mechanism. Drawing upon techniques from models like Efficient ViT [24], which reduce the computational complexity of self-attention from $O(N^2)$ to approximately $O(N)$, our ViT module adapts these cost-reduction strategies to fit within our framework.

## 4 Experiments

In this section, we introduce the two datasets utilized, the evaluation metrics, and the implementation details of our models. We train the baseline model, EGSST-B, and the extended model, EGSST-E, and compare their performance with other state-of-the-art models applied to both datasets. Detailed ablation studies are then performed to assess the impact of various components of our models. Finally, to verify the scalability of the models, we train them using varying numbers of events, obtain their corresponding weights, and analyze these weights in differently configured models.

### 4.1 Datasets and Evaluation Metrics

Two complex event camera datasets from traffic scenarios are employed in the experiments: the Gen1 Automotive Detection Dataset [46] and the 1 Megapixel Automotive Detection Dataset [9].

**The Gen1 Dataset** comprises over 39 hours of event video from urban, highway, and rural settings, with a resolution of $304 \times 240$ pixels. The dataset includes manual annotations of pedestrians and

cars, with over 255,000 labels at a frequency of 1 to 4 Hz, making it ideal for testing object detection, tracking, and optical flow algorithms in automotive environments.

**The 1Mpx Dataset** comprises over 14 hours of high-resolution (1 megapixel) event video, annotated with 25 million labels for cars, pedestrians, and two-wheelers, suitable for developing advanced detection models in dynamic traffic conditions.

**Three primary evaluation metrics** are employed in the experiments, namely the total number of parameters, the mean Average Precision (mAP@0.5:0.95) using the COCO toolbox [47], and the mean time per batch (batch size = 1) for detection. These metrics assess the models' complexity, precision, and efficiency, respectively, in real-time applications.

## 4.2 Implementation Details

The framework proposed in this study is developed using Python 3.9 and PyTorch 2.0, with graph processing powered by the advanced PyTorch Geometric library [48]. To enhance the scalability and parallelism of subgraph processing, modifications are made to the underlying libraries to achieve complete parallelization of graph processing, thus fully leveraging GPU computational capabilities. The models are trained on RTX3090 GPUs using the Lightning framework. We employ the Adam optimizer [49] coupled with the OneCycle learning rate schedule [50], which includes 100 warm-up iterations followed by cosine decay starting from the maximum learning rate. The training batch size is set at 8, with an initial learning rate of 1e-6.

## 4.3 Dynamic Label Augmentation

In our model, we use a batch segmentation method based on a fixed number of events, which requires generating corresponding labels for inputs with varying numbers of nodes. However, the data generation speed of event cameras is extremely high, while the number of labels available in existing datasets is relatively limited. For instance, current datasets such as Gen1 cannot effectively provide sufficient labels to match the rapidly collected fixed-number event data. To address this mismatch, it is necessary to develop an effective approach to increase the number of available labels.

A common approach is to extend the labeling by adding a fixed time window before and after each label, mapping all the data within that time frame to the same label [12, 27, 51, 52]. While this method can indeed increase the number of labels under typical conditions, it can introduce significant labeling errors when applied to fixed-number event-based batch segmentation, especially in dynamic environments with substantial variability.

To address this issue, we propose a dynamic label augmentation method. This approach dynamically adjusts the labeling precision based on the time span over which events are captured, aiming to expanding the number of labels while enhancing labeling accuracy. Specifically, in the dynamic label augmentation method, a longer time span for collecting a fixed number of events indicates slower target motion, allowing for an expanded time window to capture more labels for the current target. Conversely, a shorter time span suggests faster target motion, necessitating a smaller time window to maintain accurate labeling. More details can be found in the Appendix C.

## 4.4 Benchmark Comparisons

Our baseline model, EGSST-B, achieved an impressive processing time of only 2.4 milliseconds on the RTX 3090. However, this result was obtained without considering GPU power consumption. To ensure a fair comparison with other models, we conducte additional tests on the T4 GPU, which has performance comparable to the Titan Xp and RTX 1080Ti. The ASTMNet and RED models are tested on the Titan Xp, while the AEC model was evaluated on the RTX 1080Ti.

Table 1 compares the performance of various target detection methods on the Gen1 and 1 Mpx datasets. Among these benchmarks, our models EGSST-B and EGSST-E outperform the rest. EGSST-E achieves a $49.6\%$ mAP on the 1 Mpx dataset, demonstrating exceptional capabilities; EGSST-B processes in just 4.6 milliseconds on the Gen1 dataset, significantly outperforming other models in terms of efficiency and real-time processing. With parameter counts of 3.5M and 12.3M, respectively, these models achieve high performance while being more streamlined compared to traditional approaches. This optimization in parameter efficiency is particularly crucial for deployments on resource-limited platforms.

Table 1: **Comparison of Different Event-Based Vision Methods.** The results of our methods are obtained from experiments involving 10,000 events. Our methods ending in -Y utilize the YOLOX [26] detection head instead of the RT-DETR [25] method. A star * indicates that this information is not directly available and can be estimated based on modules in published articles.

| Method | Backbone | Gen1 | | 1 Mpx | | |
| --- | --- | --- | --- | --- | --- | --- |
| | | mAP (%) ↑ | Time (ms) ↓ | mAP (%) ↑ | Time (ms) ↓ | Params (M) |
| RRC-Events [32] | CNN | 30.7 | 21.5 | 34.3 | 46.4 | >100* |
| AEGNN [12] | GNN | 16.3 | - | - | - | 20.0 |
| Spiking DenseNet [20] | SNN | 18.9 | - | - | - | 8.2 |
| ERGO-12 [53] | Transformer | **50.4** | 69.9 | 40.6 | 100.0 | 59.6 |
| RED [9] | CNN + RNN | 40.0 | 16.7 | 43.0 | 39.3 | 24.1 |
| ASTMet [37] | CNN + RNN | 46.7 | 35.6 | 48.3 | 72.3 | >100* |
| AEC + DETR [54] | - | 44.5 | 7.7 | 45.9 | 20.7 | >40* |
| AEC + YOLOv5 [54] | - | 47.0 | 3.9 | 48.4 | 13.8 | >40* |
| RVT-B [27] | Transformer + RNN | 47.2 | 10.2 | 47.4 | 11.9 | 18.5 |
| GET-T [52] | Transformer + RNN | 47.9 | 16.8 | 48.4 | 18.2 | 21.9 |
| S4D-ViT-B [51] | Transformer + SSM | 46.2 | 9.4 | 46.8 | 10.9 | 16.5 |
| S5-ViT-B [51] | Transformer + SSM | 47.4 | 8.16 | 47.2 | 9.57 | 18.2 |
| **EGSST-B (ours)** | GNN + LinearViT | 44.6 | 4.6 | 45.4 | 5.1 | 3.5 |
| **EGSST-E (ours)** | GNN + LinearViT | 49.6 | 6.0 | **50.2** | 6.3 | 12.3 |
| **EGSST-B-Y (ours)** | GNN + LinearViT | 43.9 | **3.7** | 44.1 | **5.0** | 3.1 |
| **EGSST-E-Y (ours)** | GNN + LinearViT | 47.8 | 4.2 | 48.3 | 5.3 | 10.4 |

The EGSST model leverages a fully event-based architecture that makes full use of the rich spatiotemporal data generated by event cameras. The model achieves low processing latency through effective data downsampling, efficient batch processing, and the use of techniques that mimic the dynamic characteristics of the human eye. Additionally, the application of linear feature extractors further enhances its processing efficiency. These aspects enable the model to effectively utilize global information while minimizing computational demands, making it well-suited for object detection tasks using event cameras.

## 4.5 Ablation Studies

All experiments are conducted under identical environmental and equipment conditions. Given the similar performance observed on the 1Mpx and Gen1 datasets, the experiments described in this section are based on the Gen1 dataset unless otherwise stated. Moreover, since both of our models demonstrated similar test results, only the EGSST-E model is used for further analysis.

Table 2: **Impact of applying TAC.** The 'TAC Adaptive' refers to the integration of SSM with TAC, allowing for adaptive adjustments based on the data state.

| Condition | mAP (%) | Time (ms) | Params (M) |
| --- | --- | --- | --- |
| TAC Inactive | 42.1 | 4.3 | 10.2 |
| TAC Active | 51.5 | 11.1 | 16.3 |
| TAC Adaptive | 49.6 | 6.0 | 12.3 |

**The SSM and TAC**, integrated in series within our framework, are the reason why we conduct comparative experiments with these two components together. The presence of SSM dynamically activates TAC, thereby causing the parameters involved in the forward computation to vary dynamically as well. For quantitative analysis, we calculated the average amount of parameters under the condition of 10,000 event inputs across our test dataset. As shown in Table 2, the application state of TAC significantly impacts all evaluation metrics. By activating TAC at appropriate times, SSM avoids unnecessary computations and time loss, which not only enhances model efficiency but also confirms the effectiveness of SSM. Although the full application of TAC can maximize mAP accuracy, the additional time and computational costs involved are not economical.

Table 3: **Comparison after incorporating CNN.** The increase with the addition of the CNN is shown in parentheses.

| Method (with CNN) | mAP (%) | Time (ms) | Params (M) |
|---|---|---|---|
| EGSST-B | 44.6 (+0.4) | 4.6 (+0.1) | 3.51 (+0.02) |
| EGSST-E | 49.6 (+0.5) | 6.0 (+0.1) | 12.34 (+0.02) |

**Removable Convolutional Neural Network.** Our framework is event-based and does not use CNN for feature extraction. Nevertheless, it has been observed that integrating a simple CNN module enhances detection performance. To operationalize this module, a certain number of event data must be standardized across the temporal dimension to form an image-like representation. Table 3 shows that this CNN integration increases mAP with minimal impact on processing time and parameters. Consequently, we have incorporated a removable CNN layer into the model, allowing for flexibility to revert to a fully event-driven configuration if needed.

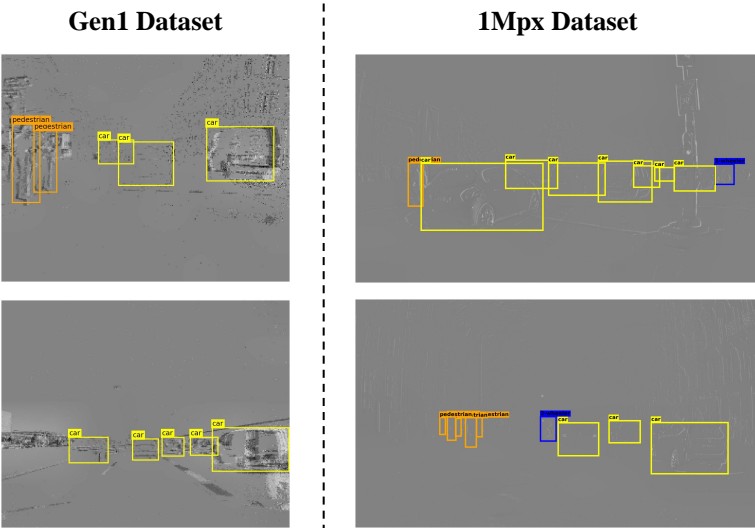

Figure 4: **Prediction Results.** Due to the low accumulated event count, the visualizations appear somewhat blurred. Nevertheless, our model effectively identifies objects within these sparse events, demonstrating its robustness and efficacy.

## 5   Discussion and Conclusion

We present a lightweight, efficient, and accurate event-based object detection framework tailored for event cameras. Our model shows strong detection performance and scalability, highlighting its practical application potential. Future work will extend this approach to object tracking and other advanced visual tasks, as well as explore deployment optimizations, such as enhancing execution speeds and converting models to ONNX or TensorRT formats. Deploying ViTs is relatively straightforward, while GNNs pose greater challenges due to complex graph processing requirements. Although progress has been made, further work is necessary to improve the deployment efficiency of GNNs.

Furthermore, we plan to investigate the integration of event data with RGB frames in a multimodal model [55, 56, 57], which could enhance the robustness and accuracy of visual recognition tasks under varied and dynamic conditions. This multimodal approach promises to leverage the strengths of both sensor types to deliver superior performance in complex environments.

## Acknowledgments and Disclosure of Funding

This research was supported by the National Key Research and Development Program under grant number 2024YFE0200703.

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

## A  Efficient ViT

This section extends section 3.5 by explaining the methodologies used in our architecture with Linear ViT, which originates from [24] and is a vital component of our model. For further details, readers are referred to the original paper.

Traditional ViTs are not suitable for fast visual tasks due to their reliance on extensive softmax attention. Softmax attention modules model interactions between every pair of tokens in the feature map to aggregate spatial information, leading to high computational complexity. Efficient ViT is a speedy visual model that replaces the computationally intense softmax calculations with multi-scale linear attention, maintaining excellent hardware efficiency while achieving a global receptive field and multi-scale learning.

Firstly, to reduce computational complexity and hardware latency, softmax attention is replaced with linear attention, formulated as:

$$O_i = \sum_{j=1}^{N} \frac{\text{Sim}(Q_i, K_j)}{\sum_{j'=1}^{N} \text{Sim}(Q_i, K_{j'})} V_j \tag{10}$$

where

$$\text{Sim}(Q, K) = \text{ReLU}(Q)\text{ReLU}(K)^T. \tag{11}$$

This leads to the final output:

$$O_i = \frac{\text{ReLU}(Q_i)\left(\sum_{j=1}^{N} \text{ReLU}(K_j)^T V_j\right)}{\text{ReLU}(Q_i)\left(\sum_{j=1}^{N} \text{ReLU}(K_j)^T\right)}, \tag{12}$$

significantly reducing both computational complexity and memory requirements.

However, due to the loss of local feature extraction capability by ReLU, linear attention's performance lags behind softmax attention. To improve performance, two core solutions have been proposed:

1. Insert deep convolutions in each FFN layer. Linear attention extracts global features, while FFN+DWConv captures local information, minimizing computational overhead and greatly enhancing local feature extraction capability of linear attention.

2. By aggregating adjacent Q, K, V token information into multi-scale tokens, the multi-scale learning capability across different channels of linear attention is enhanced.

## B   Performance Scalability Analysis

To test the scalability of our models, we use inputs with three different event counts to train distinct weight parameters and conducted cross-testing under various input conditions. Given the similar performance outcomes of EGSST-B and EGSST-E, we present specific results only for EGSST-E.

Table 4: **Performance scalability analysis with different number of input events.** The results here are all run on the Gen1 dataset and the results on 1Mpx are similar. (Note: T refers to thousand.)

| Events | EGSST-E (2T) | | EGSST-E (10T) | | EGSST-E (18T) | | EGSST-E |
| | mAP (%) ↑ | Time (ms) ↓ | mAP (%) ↑ | Time (ms) ↓ | mAP (%) ↑ | Time (ms) ↓ | Params (M) |
|---|---|---|---|---|---|---|---|
| 2,000 | 34.9 | 4.6 | 39.4 | 4.6 | 37.7 | 4.6 | 12.3 |
| 10,000 | 33.1 | 6.1 | 49.6 | 6.0 | 44.5 | 6.2 | 12.3 |
| 18,000 | 30.6 | 7.9 | 45.4 | 7.9 | 51.7 | 7.8 | 12.3 |

In terms of runtime, it can be demonstrated that a higher number of events will invariably lead to a higher consumption of time. In terms of mAP, cross-test results demonstrate that model weights trained with fewer events perform well when applied to datasets with a larger number of events, highlighting the excellent generalization capabilities of our models. Conversely, although models trained with a larger number of events show some performance disparity when applied to datasets with fewer events, the performance remains acceptable, indicating good adaptability of the model.

Our models allow for the flexible setting of the event count $N$, leveraging certain capabilities from dynamic graph processing in the treatment of GCN. While a larger event count $N$ introduces more node features, significantly enhancing detection accuracy, it also results in increased computational load and latency.

## C   Dynamic Label Augmentation

### C.1   Methodology

In this paper, we propose a novel dynamic label matching method to address the limitations of traditional label matching approaches when processing fixed numbers of event data. Specifically, the event data acquisition speed is extremely high, while existing datasets typically lack a sufficient number of labels to match the accumulated fixed-number event data. Therefore, traditional methods based on fixed time window label assignment are inadequate for scenarios requiring both adaptability to dynamic scenes and sufficient label coverage. To overcome this limitation, we design a Dynamic Label Augmentation approach to flexibly assign appropriate labels for each batch of fixed-number events.

In the proposed method, we assume that the fixed number of collected events is 10,000, for which labels need to be assigned. Suppose there are $m$ labels in the original dataset, with each label corresponding to a timestamp $\tau_m$. For each batch of 10,000 events, we first compute the representative time, defined as the mean timestamp of these events, denoted by $\bar{t}_r$, where $r$ represents the index of the current batch of 10,000 events.

To achieve dynamic label matching, we introduce a constant $\gamma$ to control the flexibility of the label matching time window. Specifically, we define a dynamically adjusted time range $\gamma \cdot \Delta t^*$, where $\Delta t^*$ is a time parameter that varies in real-time based on the dynamics of the events. A label with a timestamp $\tau_m$ is considered appropriate for the current batch of 10,000 events if it satisfies the following condition:

$$\bar{t}_r - \gamma \cdot \Delta t^* \leq \tau_m \leq \bar{t}_r + \gamma \cdot \Delta t^* \tag{13}$$

Since $\Delta t^*$ changes dynamically according to the characteristics of the events, the proposed method can effectively adapt to variations in the event data, ensuring the accuracy of label assignment.

In summary, the Dynamic Label Augmentation method enables flexible adjustment of the time window, allowing the label assignment process to adapt to different dynamic properties of the events. This approach avoids potential mismatches associated with traditional fixed time window methods and exhibits strong robustness and adaptability.

## C.2 Experiment

Table 5 illustrates the impact of various data augmentation techniques on model accuracy, with dynamic label augmentation demonstrating a relative advantage. Traditional augmentation methods, such as horizontal flipping, zooming in, and zooming out, improve model robustness by increasing data diversity, contributing to accuracy improvements. However, dynamic label augmentation, which adaptively adjusts the label generation range, shows better adaptability when handling dynamic scenes, particularly by reducing the risk of label mismatches, thereby potentially enhancing overall model performance.

Table 5: **Accuracy Improvement from Dynamic Label Augmentation.** All augmentation techniques improve accuracy, with dynamic augmentation showing the greatest improvement.

| h-flip | zoom-in | zoom-out | dynamic | mAP (%) |
|--------|---------|----------|---------|---------|
|        |         |          |         | 38.3    |
| ✓      |         |          |         | 41.8    |
|        | ✓       |          |         | 43.3    |
|        |         | ✓        |         | 42.1    |
|        |         |          | ✓       | 46.6    |
| ✓      | ✓       | ✓        | ✓       | 49.6    |

The experimental results suggest that while conventional augmentation techniques are effective in improving accuracy, dynamic label augmentation may provide additional gains in both accuracy and robustness by flexibly adjusting the label generation process. This approach appears to improve the model's ability to adapt to complex and fast-changing environments to a certain extent.

