# OpenReview forum: "EGSST: Event-based Graph Spatiotemporal Sensitive Transformer for Object Detection"
_NeurIPS.cc/2024/Conference — NeurIPS 2024 poster_

### Official Review · Reviewer_NBbr · 2024-06-27

**Soundness:** 2
**Presentation:** 2
**Contribution:** 3
**Rating:** 5
**Confidence:** 3

**Summary:**

This paper introduces a novel event-based object detection network by processing event-based data as graph data. After incorporating an SSM that acts as the selection role, graph convolution neural networks, and various attention modules, the proposed pipeline achieves good results while retaining high efficiency. Ablation studies prove the effectiveness of the proposed modules. Although the overall results are outstanding, the overall writing is poor, making the paper hard to follow.

**Strengths:**

1. Processing event-based data as graph data to retain its spatiotemporal information is an interesting direction worth exploring.
2. The novel design of the SSM module suits the property of event-based data well.
3. The experiments illustrate the effectiveness and efficiency of the proposed method.

**Weaknesses:**

1. In section 3.3, the author claims that the SSM module mimics the human eye to output the degree of object dynamics in the event stream. This metric is measured by the number of events and the time span of each subgraph since more events usually indicate faster object movement. However, this measurement does not consider huge objects in the static background which also generate a large number of events with relatively low speed, which does not align well with the author's claim. Also, the definition of $f(\cdot)$ of SSM is not clear, which is only given by the author in line 176 as an example.

2. The $\pi$ function in section 3.3 is not detailedly introduced either.

3. In line 302, the author mentions that the CNN module is removable. Where is this removable CNN module? Additionally, the TAC module, Linear ViT, and DETR all process frame-based data, and CNN is included in the Linear ViT as illustrated in Figure 2. Based on these observations, I don't think the framework is event-based.

**Questions:**

1. In line 294, the author indicates that the parameters involved in the forward computation vary dynamically due to the SSM module. Could the authors provide some statistical and qualitative results on the ratio of involved parameters to help readers better understand the role of SSM?

2. How is the gating signal produced from $F_{st}$ (line 215)? Could the author provide experimental results to prove the effectiveness of generating $Q$ from $F_{st}$ instead of $\mathcal{F}$?

3. In this paper, only quantitative results are shown, making me doubt whether I am reading a paper in the CV field, especially for the object detection task. Could the authors show more qualitative results? I think this is more valuable and significant than the data augmentation results which can be moved to the appendix.

**Limitations:**

The authors are very transparent about their limitations on deploying the GNN into production.

---

> ### Author Rebuttal · Authors · 2024-08-07
>
> Thank you for your constructive feedback. We have carefully revised the manuscript based on your suggestions, particularly regarding the clarity of function definitions and the roles of specific modules. We hope these revisions meet your expectations and ask for your reconsideration during the review process. We are committed to addressing any further questions or requirements during the discussion period.
>
> **W1.** In response to the first issue you raised in the weaknesses section, we provide the following detailed replies and corrections:
>
> 1) To further illustrate the relationship between object and background dynamics, we have added an image of SSM in the rebuttal PDF, positioned near section 3.3 of the manuscript. Regarding the dynamics, large objects generate a large number of events, but their relatively slow movement speed results in a large time span. These events significantly influence the computation of the Event Global Motion (EGM), effectively reflecting the overall scene dynamics. Conversely, fast-moving objects, while generating a similar number of events, have a shorter time span, leading to a higher Event Local Motion (ELM).
>
> 2) We originally designed $f(\cdot)$ to quantify the dynamics in the event stream using the number of events and time intervals. However, our experiments indicate that a simple proportional function $f(x, y) = \frac{x}{y}$ adequately captures these dynamics without needing additional parameters from neural networks. To clarify and prevent misunderstandings, we will refine our explanation of this function in the revised manuscript, moving beyond examples to a clear definition.
>
> **W2.** Thank you for highlighting the unclear definition of function $\pi$ in our manuscript. To address this, we will enhance our description in the revised paper. Function $\pi$ performs aggregation operations—minimum, maximum, and mean—to enable lightweight processing with flexibility. In our setup, we primarily use the mean method to compute centroid coordinates of input data points. Our findings suggest that the choice of aggregation method does not significantly affect the overall detection performance. We will detail this analysis in the revised manuscript, providing ablation study results to clarify the effectiveness of function $\pi$.
>
> **W3.** Thank you for your comments.
> 1) To avoid any confusion about the CNN module, we will clearly mark this removable component in the updated Figure 2.
> 2) Regarding data processing being event-based or frame-based: Our method incorporates components from established object detection frameworks like YOLO and DETR but differs fundamentally in data processing and feature extraction. Initially, our feature extraction relies solely on graph structures to process event data, preserving rich spatiotemporal information unlike traditional methods that compress events into frames. Secondly, when converting graph features to frame features, we implement event-level operations. By mapping each event's features to specific spatial locations, we ensure that every position within the frame contains detailed event-level information. This approach maintains the integrity of the event data throughout the frame-based output, significantly reducing the information loss typical of standard frame compression techniques.
>
> **Q1.** In response to the first question, we recognize the need for more detailed explanations in our manuscript and offer the following clarifications:
>
> 1) Experimental Setup: We used a fixed batch size of 1 across 30,000 batches, activating the TAC module for about 34.4% of the total event data. These details will be included in section 4.4 to better illustrate the dynamics of the TAC module during operations.
>
> 2) Correction of Errors: We identified errors in our previous ablation studies setup where the shuffle option was incorrectly active, and the batch size was not consistently 1. This affected the numerical results in the TAC Inactive line, and the corrected mAP should be 42.1%, with processing time of 4.3 ms. These corrections will be detailed in the revised manuscript.
>
> **Q2.** Thank you for your attention to the details of our research methodology.
>
> 1) Flowchart Supplement: We have detailed the process flowchart of the TAC module in the revised Figure 2, which can be viewed in the PDF file.
>
> 2) Ablation Study Results: To verify the effectiveness of generating Q from Fst, we designed an ablation study to compare the outcomes of generating Q from Fst and from F. The results show that the method of generating Q from Fst with TAC always active increased the mAP accuracy by 0.3%, while the time delay increased by only 0.01 ms and did not lead to any additional parameter increase. Although there is a slight time delay, considering that the improvement in accuracy is more significant compared to the minimal time delay, we believe that obtaining Q from Fst is an effective and worthwhile method.
>
> **Q3.** Thank you for your feedback. Based on your suggestion to include more qualitative results, we have implemented the following improvements:
> 1) SSM Visualization: We've added visualizations of the SSM module's output to the PDF, providing a clear view of object dynamics within scenes.
> 2) Qualitative Detection Results: We've incorporated several images showing detection outcomes in various real-world settings, highlighting our model's effectiveness in complex environments.
> 3) Data Augmentation Details: As suggested, we will move the data augmentation results to the appendix, focusing the main text more on our key contributions.
>
> **Limitations：**
>
> Deploying GNNs into production involves addressing practical challenges such as computational resources, inference performance, and data storage and management, requiring concerted efforts in both hardware and software.

---

> > ### Comment · Reviewer_NBbr · 2024-08-11
> >
> > Dear authors,
> >
> > Thanks for your response. Regarding W1, say there are two objects with the same speed, where one object is larger than the other one. In this condition, would the relative value of the larger object be larger than the other object? Then would the model neglect the relatively small object due to the smaller relative value?  I think the provided qualitative results are not enough to resolve my concern since the truck on the right side in Figure 2b has a larger N and smaller $\Delta t$, deserving a larger relative value. Instead, I want to see some comparisons between two objects one of which has a larger shape and larger $\Delta  t$. Since the rebuttal file can't be renewed, I hope the author can explain this condition in words more detailedly.
> >
> > Also, if I didn't miss something, it seems that you didn't answer the first part of my question 2, i.e., how is the gating signal produced from $F_{st}$ (line 215)?

---

> > > ### Author Response · Authors · 2024-08-13
> > >
> > > Thank you very much for your detailed examination of our method. Here are our responses to your questions:
> > >
> > > 1. Regarding the issue you mentioned about large and small objects moving at the same speed but having different relative values: We have largely mitigated this issue when constructing connected subgraphs by introducing inter-subgraph attention to capture relationships across subgraphs. Specifically, by setting a distance threshold $R$ and limiting the number of neighboring nodes, a single large object may consist of multiple connected subgraphs, and the larger the object, the more subgraphs it includes. Thus, the relative value of the same object is determined by the combined relative values of these subgraphs. Since multiple connected subgraphs can segment the integrity of the target, we have subsequently introduced an inter-subgraph Graph Attention Network (GAT) (in line 202), which strengthens the connections within subgraphs of the same object and weakens those between different objects. Additionally, the GAT does not overlook relatively smaller objects; instead, it adaptively assesses their importance within the overall context.
> > >
> > > 2. We have included a detailed flowchart of the Temporal Activation Controller (TAC) in Figure 1 of the rebuttal PDF. After obtaining $F_{st}$ through convolutional aggregation, it is processed by a combination of two well-designed convolutional neural networks, the Attention Network and the Gating Network, to determine the probability values (i.e., gating values) for each Feature Map. Finally, these values are fused within the Fusion module with the original Feature Maps.

---

> > > > ### Comment · Reviewer_NBbr · 2024-08-13
> > > >
> > > > Thanks for your further clarification.
> > > >
> > > > Regarding the first point, I think the author should give a more detailed introduction to the construction of subgraphs in the main paper and give some experiments to explain this condition. This should be done in the revised version.
> > > >
> > > > Overall, this paper lacks some essential explanations of its method and does not contain qualitative results. The provided qualitative results in the rebuttal file do not contain comparisons with other methods, as well as an explanation of the color of bounding boxes. The presentation is not good enough.
> > > >
> > > > However, the experimental results are good at both performance and cost time, and the authors chose to release their code, which would be beneficial to the community. I would like to raise my rating to borderline accept, and I hope the authors carefully revise their paper for better understanding and soundness.

---

> > > > > ### Author Response · Authors · 2024-08-14
> > > > >
> > > > > We sincerely appreciate your detailed comments on our work. We will carefully organize the discussed points and incorporate them into our revised version. Specifically, we will provide detailed explanations on subgraph construction, dynamics, intra-subgraph and inter-subgraph processing, and their interrelationships in Section 3. Regarding the issue you raised about the insufficient quality of qualitative results in the rebuttal file, we have been working on improvements, and the final visual results will be more refined and polished.
> > > > >
> > > > > We believe that, with your help and the feedback from other reviewers, we can present our paper more comprehensively and clearly. We sincerely thank you again for your valuable insights.

---

### Official Review · Reviewer_NWmb · 2024-07-10

**Soundness:** 3
**Presentation:** 3
**Contribution:** 3
**Rating:** 6
**Confidence:** 5

**Summary:**

The paper proposes a novel event-based graph spatiotemporal sensitive network (EGSST), which is the first work using Graph Transformer for object detection tasks on event cameras. This work primarily involves two key innovative modules: a spatiotemporal sensitivity module (SSM) and an adaptive temporal activation controller (TAC). Additionally, the integration of a lightweight, multi-scale Linear Vision Transformer (LViT) significantly enhances processing efficiency. The results demonstrate that the proposed EGSST achieves state-of-the-art (SOTA) performance, especially outperforming other graph neural networks for event-based object detection.

**Strengths:**

i) Graph neural networks for event-based object detection is a promising solution to achieve low-latency in real-world applications.

ii) The proposed EGSST achieves SOTA performance, especially outperforming other GNNs for event-based object detection.

iii) The writing is straightforward, clear, and easy to understand.

**Weaknesses:**

i) The authors claim that the first Graph Transformer worked on event-based object detection, seemingly simply linking the feature maps generated by GNNs directly to DETR. Of course, I am certain that the author has contributed to the handling of events in spatiotemporal GNNs. However, the author may not make any innovative work on the unique combination of GNNs and Transformer. Could the author clarify the innovative aspects of the combination of graph transformer and others?

ii) In Table 1, the authors should separately place the existing method of event-based object detection using GNNs in the bottom few rows of the table. There are still several works that have not been searched and cited, such as DAGr in Nature 2024. In addition, by comparing the GNNs method, it can be found that the proposed EGSST has good performance and inference time.

iii) The authors' use of DETR to regress the output of object detection results is not consistent with the YOLOX approach used by existing GNNs for event-based object detection. It is challenging to determine whether the DETR module has significantly improved performance. The authors should conduct an experiment with a YOLOX detection head to clarify this.

iv) In the ablation experiment, the authors should ablate important parameters in the proposed innovative modules (i.e., SSM and TAC). In addition, the expansion experiment on the accumulation of event windows in Table 5 takes up too much space, which may affect the length of other important experimental content.

v) The writing section needs further improvement. For example:

a. In related works, the author can write according to event-based object detection, graph neural networks for event data. This writing may better reflect the focus and novelty of the work.

b. The biggest advantage of GNNs in processing event data is low latency. The author proposes a method that basically reaches the millimeter level, which is very good. It is recommended to emphasize the low latency advantage of the method more.

**Questions:**

Please see the weakness and response each comment. Besides, Additionally, the authors should also answer the following question: Could the authors plan to further expand this unimodal work to multimodal work by integrating GNN for event processing and Transformer for image processing in the future?

If so, please cite some multimodal object detection methods [1, 2, 3] using events and frames.

[1] Event-based vision enhanced: A joint detection framework in autonomous driving, ICME 2019.

[2] SODFormer: Streaming object detection with transformer using events and frames, TPAMI 2023.

[3] Low-latency automotive vision with event cameras, Nature 2024.

**Limitations:**

A minor flaw is that the authors claim to be the first to apply Graph Transformer to event-based object detection, seemingly by merely linking feature maps generated by GNNs directly to DETR. It suggests that the authors revise a statement to be slightly more conservative.

---

> ### Author Rebuttal · Authors · 2024-08-07
>
> **W1.** We greatly appreciate your evaluation and suggestions regarding our integration of Graph and Transformer technologies. In our integrated framework, there are two key aspects:
>
> 1) Interaction Mechanism between SSM and TAC: The Spatiotemporal Sensitivity Module (SSM) we designed processes event data based on the graph structure and directly influences the activation state of the Temporal Activation Controller (TAC). This design enables the TAC to precisely focus on the temporal dynamics of the event data, thereby enhancing processing efficiency and response speed.
>
> 2) Event-level Information Transformation: In converting graph features to the format required by Transformers, we employ event-level operations. By accumulating features of each event at their corresponding spatial locations, each location not only retains rich event information but also reflects underlying dynamic changes, which is critical for object detection in dynamic vision systems.
>
> Of course, we acknowledge the shortcomings pointed out by you in this section and in the later Limitations. Therefore, we have attempted to make our statements about the Graph Transformer more conservative, as follows:
>
> - "Our model effectively combines the advantages of Graph and Transformer technologies. Based on event data, it is lightweight, fast, and accurate, providing a novel technological approach for performing object detection tasks within event data."
>
>
>
> **W2.** Thank you very much for your careful reading and valuable suggestions. We will reorganize and optimize the method ordering in Table 1 according to your suggestions and will include some of the latest related literature, including S4D-ViT-B[1], S5-ViT-B[1], GET-T[2], and ERGO-12[3], to ensure our paper covers the most current research advancements.
>
> Furthermore, based on your recommendation [4], we learned about the DAGr method, an innovative hybrid approach that combines event cameras with traditional frame cameras. DAGr leverages the high temporal resolution and sparsity benefits of event cameras, along with the rich contextual information from frame cameras, to achieve efficient, fast object detection while significantly reducing perception and computation latencies. Unfortunately, the DAGr method was published online just days after the NeurIPS 2024 submission deadline, which is why we were unable to reference this excellent work in time. However, we are impressed by this method and plan to use it as an important benchmark in our future multimodal research.
>
> References:
>
> [1] State Space Models for Event Cameras, CVPR 2024
>
> [2] Get: Group event transformer for event-based vision, ICCV 2023
>
> [3] From chaos comes order: Ordering event representations for object recognition and detection, ICCV 2023.
>
> [4] Low-latency automotive vision with event cameras, Nature 2024.
>
>
> **W3.** Thank you for your detailed attention to our choice of detection head technology. In response to your concerns about using the DETR or YOLO series detection heads, we have added additional experimental results using YOLOX as the detection head. These new results will be included in Table 1 to visually demonstrate the performance of YOLOX within our framework.
>
> Due to the word count and PDF page limitations of the rebuttal, we are unable to detail all of the revisions and comparisons here. Therefore, only the additions in Table 1 are given in the author rebuttal.
>
>
>
> **W4.** Thank you for your detailed guidance on our ablation study design and the layout of our paper. To evaluate the impact of SSM and TAC modules, we add ablation experiments such as $\pi$ in SSM and Fst in TAC. The experimental results indicate that while $\pi$ has a negligible overall effect, generating Q from Fst with TAC always active improves precision, increasing mAP accuracy by 0.3% with only a 0.01 ms increase in time delay and no additional parameter increase. More detailed ablation experiments will be added in the Appendix. Additionally, we have included visualizations of SSM in dynamic environments, further confirming the effectiveness of these combined modules. These results will help readers more intuitively understand the roles of the modules.
> Regarding your concern about the space occupied by the extended experiments on event window accumulation, we have recognized this issue and plan to move this section to the appendix in the revised manuscript to save space in the main text and maintain the compactness of the paper.
>
> **W5.** Thank you very much for your specific suggestions regarding our writing. We will carefully revise and improve our paper based on your feedback.
>
> 1) Related Work Section: We will reorganize the "Related Work" section to more detailedly categorize and discuss it according to event-driven object detection and graph neural network processing of event data, in order to better highlight the focus and innovations of our work.
>
> 2) Emphasizing Low Latency Advantage: We will more clearly emphasize in the paper the achievements of our method in achieving millisecond-level latency.
>
> Due to word count limitations, we will reserve more detailed modifications and explanations for the revised manuscript. We hope these improvements will fully meet your expectations and further enhance the quality of the paper. Thank you for your valuable suggestions, and we look forward to you seeing these changes in the revised manuscript.
>
> **Questions:**
>
> Thank you for your question. Indeed, in future research, we plan to expand our current unimodal work to multimodal efforts. We believe that by integrating data from event cameras and traditional frame cameras, we can further enhance the system's perceptual capabilities and response speed, especially in dynamic and complex visual environments.
>
>  To support our future research direction and acknowledge the existing work in our field, we will cite the excellent studies you mentioned in the "Discussion and Conclusion" section of our paper.

---

> > ### Comment · Reviewer_NWmb · 2024-08-13
> >
> > The author has addressed my concerns, and I will maintain the original score. I hope the author will incorporate the promised modifications into the camera-ready version. Additionally, I encourage the authors to explore several multimodal object detection methods and refer to the multimodal literature I recommended for future work.

---

> ### Comment · Area_Chair_AW4a · 2024-08-12
>
> Dear Reviewer
>
> The author-discussion period ends at August 13, which is just 1 day away. Can you please discuss the rebuttal and the paper with the authors? Was there any concern that the authors' rebuttal did not address? Do you need further clarification from the authors?
>
> Best Regards,
>
> Your AC

---

### Official Review · Reviewer_WqtW · 2024-07-12

**Soundness:** 2
**Presentation:** 2
**Contribution:** 3
**Rating:** 5
**Confidence:** 5

**Summary:**

This paper uses graph structure to model event data and realize event classification. Spatiotemporal Sensitivity Module (SSM) and an adaptive Temporal Activation Controller (TAC) mimic the response of the human eyes in dynamic environments by selectively
activating the temporal attention mechanism based on the relative dynamics of event data, thereby effectively conserving computational resources. In addition, the integration of a lightweight, multi-scale Linear Vision Transformer (LViT) markedly enhances processing efficiency.

**Strengths:**

In this paper, SSM and TAC, as well as multi-scale Linear Vision Transformer (LViT) are used to improve the efficiency of calculation while maintaining good classification performance.
The data augmentation method of Dynamic Label Augmentation is also mentioned as a potential contribution, but is lacking in specifics.

**Weaknesses:**

1. Insufficient experiment. The Gen1 and 1Mpx datasets used in this paper are not the most widely used datasets for event classification tasks. Even with a focus on driving scenarios, the N-CARS [HATS: Histograms of averaged time surfaces for robust event-based object classification] dataset should be included for comparison with other existing methods.
2. The description of some details needs to be perfected. For example, the specific process of Dynamic Label Augmentation (312 lines) was not introduced, the key parameter $R$used in the experiment (137 lines, formula 2) was not clear, and the main idea of Efficient ViT (236 lines) used in the paper should also be briefly explained.

**Questions:**

The highlight of this paper is the improvement of efficiency. I want to know whether the time statistics of the algorithm include the Graph Construction process, and how to calculate the time of the algorithm? What percentage of the total process of the algorithm is occupied by the construction time? How do the relevant hyperparameters (such as the number of edges, neighborhood radius, etc.) affect the algorithm when used?

**Limitations:**

In addition to the limitations mentioned in Weaknesses, do the methods mentioned in this paper generalize to other visual tasks such as optical flow estimation? Verifying the generalization of the method proposed in this article will make the research more meaningful.

---

> ### Author Rebuttal · Authors · 2024-08-07
>
> **W1.** Thank you for your attention to our choice of datasets. We acknowledge that N-CARS [7] is a significant dataset for event classification tasks. However, our research focuses on event-based object detection, which is distinct from event classification. Literature [1,2,3,4] supports our use of the Gen1 and 1Mpx datasets, which are well-established in object detection research, hence their selection for our studies. Nevertheless, we recognize the potential of the N-CARS dataset for future research in event classification. We plan to explore this dataset further and will discuss these plans in our paper, citing relevant literature to support our research direction and dataset choices.
>
> **W2.** Thank you very much for highlighting areas for improvement in our paper. We have provided more detailed explanations regarding Dynamic Label Augmentation, the key parameter R, and the Efficient ViT as follows:
>
> 1. **Dynamic Label Augmentation:** We will detail the methodology in the appendix and summarize it here. This technique dynamically adjusts the size of label expansion windows based on the time span of accumulated events, enhancing label accuracy. Unlike traditional methods that use fixed ranges—potentially leading to errors in dynamic scenes—our method adapts the expansion windows to suit the speed of moving objects, minimizing incorrect labeling. For example, expansion windows contract for fast-moving objects to avoid mislabeling, and expand in slower-changing scenes to capture more potential labels. This adaptive approach is particularly effective in environments with variable dynamics, such as traffic monitoring or motion tracking, ensuring objects are labeled more precisely.
> 2. **Parameter R:** We value your interest in the role of parameter R, a key hyperparameter determining the distance threshold for edges between graph nodes. We set R to 30 across two datasets, a value optimized through initial experiments to effectively construct graph data and generate adequate connected subgraphs for our analysis. We further enhance graph construction efficiency using the radius_graph function from the Pytorch Geometric library, which rapidly builds graphs based on this radius, thereby improving our model’s processing speed and efficiency.
> 3. **Efficient ViT [5]:** We appreciate your interest in Efficient ViT. For comprehensive details, please see its original ICCV 2023 publication. In our appendix, we will outline the core design and technical merits of Efficient ViT for rapid visual tasks:
>    - **Softmax Replacement:** Traditional ViT models use softmax attention, which is effective but computationally intensive. Efficient ViT uses multi-scale linear attention to reduce computational complexity and hardware latency.
>      - $ O_i = \sum_{j=1}^N \frac{\text{Sim}(Q_i, K_j)}{\sum_{j=1}^N \text{Sim}(Q_i, K_{j})} V_j  $
>      - $ \text{Sim}(Q, K) = \text{ReLU}(Q) \text{ReLU}(K)^T $
>      - Consequently, $O_i=\frac{\text{ReLU}(Q_i) \left(\sum_{j=1}^N \text{ReLU}(K_j) V_j\right)}{\text{ReLU}(Q_i) \left(\sum_{j=1}^N \text{ReLU}(K_j)\right)^T},$ drastically reducing computational complexity and memory demands.
>    - **Optimizing Local Feature Extraction:** Linear attention reduces computational demands but is less effective at capturing local details. We address this by including depthwise convolution (DWConv) in each Feed Forward Network (FFN) layer.
>    - **Aggregating Multi-Scale Token Information:** The model aggregates neighboring Q, K, and V tokens into multi-scale tokens, enhancing linear attention's ability to process data across different channels efficiently and accurately.
>
> **Questions:**
>
> **Re.** Thank you for your detailed inquiry about our model. We are happy to provide clarifications:
>
> 1) Time Statistics: Our algorithm's timing involves the graph processing module, the Linear ViT module, and the detection head module. Graph processing includes constructing the graph and operations within the Graph Neural Network (GNN) and Spatiotemporal Sensitivity Module (SSM), with precise GPU timing provided by PyTorch's torch.cuda.Event.
>
> 2) Time Proportion: The graph processing module, utilizing the Pytorch Geometric library and the torch_scatter library for efficient sparse operations, is highly efficient, accounting for approximately 31.4% of the total algorithm time. This efficiency significantly boosts our model's performance in dynamic environments.
>
> **Limitations:**
>
> Thank you for highlighting the potential applications and emphasizing the need for broader validation of our method. Currently, our research focuses on event-based object detection, allowing us to refine detection techniques specific to event data.
>
> However, we understand the importance of extending these techniques to a wider range of visual tasks. In future work, we plan to test the model's applicability to other areas, including event classification [1, 2, 4, 6] and optical flow estimation. We will continue our research to evaluate the generalization and adaptability of these methods and report our findings. We look forward to our continued contributions to both the research and practical applications in event camera vision.
>
> **References:**
>
> [1] Learning to detect objects with a 1 megapixel event camera, Advances in Neural Information Processing Systems, 2020.
>
> [2] Recurrent vision transformers for object detection with event cameras, CVPR 2022.
>
> [3] Asynchronous spatio-temporal memory network for continuous event-based object detection, IEEE Transactions on Image Processing, 2022.
>
> [4] Better and faster: Adaptive event conversion for event-based object detection, AAAI 2023.
>
> [5] Efficientvit: Lightweight multi-scale attention for high-resolution dense prediction, ICCV 2023.
>
> [6] Aegnn: Asynchronous event-based graph neural networks, CVPR 2022.
>
> [7] HATS: Histograms of averaged time surfaces for robust event-based object classification, CVPR 2018.

---

> > ### Comment · Reviewer_WqtW · 2024-08-13
> > **Unresolved and new questions**
> >
> > Your response partially addressed my question. However, since a significant portion of the article focuses on graph construction, I'm still more interested in that aspect.
> >
> > Q1: The parameter $R$ is related to the normalization factor $β$ mentioned in line 133. I believe that both $β$ and the $R$ parameter significantly impact the subsequent results. How is  $β$ determined, and what is the analysis of the relationship between $β$ and $R$ (particularly concerning different datasets or motion scenarios)?
> >
> > You mentioned that $R=30$. Could you explain how this value was set? Additionally, could you provide a brief overview of the setting strategy and any comparisons made?
> >
> > Q2: What is the specific purpose of Equations 6 and 7? I don't seem to find where they are used in the subsequent sections.
> >
> > Q3: As raised by reviewer NBbr, in line 195, $π$ is the function used to perform the aggregation of the coordinate sets. This aggregation function needs at least a brief introduction, as you mentioned in your response to reviewer NBbr.
> > Following up on Q1 and Q2, my concern is to what extent the aggregation function reduces the subsequent data processing load. This will help me understand whether the efficiency improvements of your proposed method are primarily focused on the initial graph construction or on the subsequent graph network processing. To what extent can the efficiency be improved through optimization after constructing connected subgraphs and the aggregation of nearby events? Ablation studies on the impact of hyperparameter settings during the graph construction process on both efficiency and accuracy should be considered.
> >
> > Q4: Does the provided code include the section for constructing connected subgraphs? I recommend adding a ReadMe file in the subsequent code versions to facilitate quicker access to relevant content.

---

> > > ### Author Response · Authors · 2024-08-13
> > >
> > > Thank you very much for the series of questions you have raised. We will address each of these questions in turn.
> > >
> > > **Q1.** Regarding the setting of $\beta$, we referred to the configurations described in references [1] and [2], aiming to normalize temporal locations via a factor and map them to a range similar to that of spatial coordinates.
> > >
> > > As our method accumulates a fixed number of events and the model is event-based, there is no need to consider the impact of time spans on imaging like frame-based models. As long as the number of events accumulated by the model is consistent and the normalization range of temporal locations across datasets is identical, the selection of $R$ can naturally be consistent.
> > >
> > > The determination of the parameter $R$ was made during the construction of the graph, taking into account the number of subgraphs and the number of nodes within each subgraph. Specifically, our setup retains about two-thirds of the total number of nodes, making the number of subgraphs approximately ten times the total number of objects. $R=30$ has been sufficient to enable the model to effectively focus on relevant objects, achieving low processing latencies and high detection accuracies.
> > >
> > > **Q2.** Equations 6 and 7 are formulas for discriminating event dynamics. The results of these equations are incorporated as part of the features into the model (in line 197), influencing the dynamic indicators generated subsequently.
> > >
> > > **Q3.** Regarding the suggestions made by you and reviewer NBbr, we will provide a brief introduction to the aggregation function $\pi$ and incorporate it into the revised version.
> > >
> > > Regarding your question about how the aggregation function alleviates the burden of subsequent data processing and where our method primarily improves efficiency, we offer the following explanations:
> > >
> > > - The aggregation function you are concerned with can essentially be understood as a graph pooling operation. It pools a large number of nodes and features from each subgraph into a few representative features, which are comprehensive summaries of all node features within each subgraph. Pooling is a common technique in graph processing, and our method adheres to standard pooling practices. For more details on different types of graph pooling operations and the potential efficiency gains they offer, please refer to reference [3]. We consider graph pooling to be a consensus method, hence no further testing experiments were conducted.
> > > - In terms of efficiency improvements, our model's design aims to fully utilize the spatiotemporal characteristics of event data, minimize redundant computations, enhance efficiency, and reduce latency. In the graph processing component, we filter out unnecessary event nodes (including significant noise) through the construction of connected subgraphs and related graph processing operations, thereby enhancing the model's focus on effective objects and significantly reducing the computational load of graph processing due to the decrease in the number of nodes. Operations that reduce spatiotemporal processing and enhance processing efficiency are primarily focused on the combined application of SSM, TAC, and Linear ViT, with detailed descriptions of ablation studies included in the manuscript. For explanations regarding the settings of hyperparameters, please refer to the response to Q1.
> > >
> > > **Q4.** The provided code includes the part for constructing connected subgraphs. We will update the code to add detailed comments and a ReadMe file for readers' reference.
> > >
> > >
> > > **References:**
> > >
> > > [1] Aegnn: Asynchronous event-based graph neural networks, CVPR 2022.
> > >
> > > [2] Graph-based object classification for neuromorphic vision sensing, ICCV 2019.
> > >
> > > [3] Graph neural networks: A review of methods and applications, AI Open 2020.
> > >
> > >
> > > If our response has addressed your concerns, we kindly ask you to reconsider your rating. If you have any further questions, we are more than happy to address them.

---

> > > > ### Comment · Reviewer_WqtW · 2024-08-14
> > > >
> > > > Based on your response, I still have the following questions.
> > > >
> > > > Q1: The approach of handling a fixed number of events can indeed reduce the impact of varying motion scenarios, but differences in the proportion of moving objects within a scene can still affect the event generation rate. Should $R$, $β$ and the number of connected subgraphs in this paper be considered as hyperparameters that need adjustment?
> > > >
> > > > Q2: The description in line 197 is not sufficiently clear. Please make sure to refine it in the next revision.
> > > >
> > > > Q3: Is the method of constructing connected subgraphs similar to a clustering process, thereby reducing the impact of noise points? The approach of using graph-cut algorithms and clustering in the context of event cameras has already been proposed. Therefore, the contribution of the "use of connected subgraphs in the context of event cameras" may not be solid. Examples include "Event-based Agile Object Catching with a Quadrupedal Robot" and "Event-based Motion Segmentation with Spatio-Temporal Graph Cuts."
> > > >
> > > > It is crucial to experimentally demonstrate the efficiency gains and impact on performance brought by the graph construction process compared to the graph network processing.
> > > >
> > > > Q4: I wasn't able to find the graph construction and optimization components based on the file names in the provided code. Could you please indicate the relevant code path so I can better understand your work through the code?

---

> > > > > ### Author Response · Authors · 2024-08-14
> > > > >
> > > > > Thank you very much for raising these further concerns. We will do our best to address them:
> > > > >
> > > > > **Q1.**
> > > > >
> > > > > Your understanding is correct, but we would like to emphasize a few additional points. The difference in the proportion of moving objects does indeed affect the event generation rate, which in turn influences the model. To mitigate this impact, and to better capture relationships between objects of varying proportions, we have incorporated an inter-subgraph attention mechanism through the Graph Attention Network (GAT) module. This module adaptively strengthens or weakens the connections across subgraphs.
> > > > >
> > > > > While $R$, $β$, and the number of connected subgraphs can indeed be considered hyperparameters that may require adjustment, the addition of the GAT module has greatly enhanced the model’s robustness, providing it with greater adaptability to varying data.
> > > > >
> > > > >
> > > > >
> > > > > **Q2.**
> > > > >
> > > > > Regarding the issue you raised about the unclear description in line 197, we will address it in the next revision by incorporating both your suggestions and those of reviewer NBbr. We plan to provide a more detailed explanation in Section 3.3, elaborating on the construction of subgraphs, their dynamics, and the relationships between them. The key points discussed with the reviewers will be included to enhance the clarity of the description.
> > > > >
> > > > >
> > > > >
> > > > > **Q3.**
> > > > >
> > > > > The method of constructing connected subgraphs is indeed similar to graph clustering in terms of its effect, but this is only one part of our preprocessing steps. The construction of connected subgraphs helps to reduce the impact of noise points, but it is just one component of our overall graph processing approach. Subsequent steps include operations within the subgraphs, processing across subgraphs, and so on, which together form our model's processing method at the graph level.
> > > > >
> > > > >
> > > > >
> > > > > **Q4.**
> > > > >
> > > > > We sincerely apologize for the issue with the code. This part of the code was supposed to be included in graph_handling.py. Initially, we believed that the complete code had been uploaded, but upon further inspection, we found that the specific part of the code you needed was indeed missing. Since the supplementary materials cannot be updated at this time, we will provide the complete code and accompanying documentation in the future.
> > > > >
> > > > > We deeply apologize once again for any inconvenience caused by our oversight. Therefore, we would like to briefly explain the relevant parts of the code here. For graph construction, we used the radius_graph function from PYG (PyTorch Geometric). For subgraph construction, we employed the LargestConnectedComponents class from PYG. The subsequent graph processing operations are based on the PYG library and the torch_scatter library, enabling efficient graph processing on the GPU.

---

> > > > > > ### Comment · Reviewer_WqtW · 2024-08-14
> > > > > >
> > > > > > Regarding the impact of parameters on effectiveness and efficiency discussed in Q1, the authors should include a discussion in the appendix of the final version of the paper. Similarly, additional experiments should be added in response to Q3 to clarify the contributions of the graph construction optimization process and subsequent graph network processing to both efficiency and effectiveness. Additionally, the publicly available code should be promptly updated.
> > > > > >
> > > > > >
> > > > > > Overall, the paper's presentation is not very clear, and its novelty is modest. However, the method does show some improvements in effectiveness and efficiency and provides corresponding code. If the authors can thoroughly address the reviewers' comments, I would consider raising my rating to borderline accept.

---

> > > > > > > ### Author Response · Authors · 2024-08-14
> > > > > > >
> > > > > > > We will do our best to address the issues you raised and reflect them in the final revised version. We sincerely appreciate your detailed comments on our paper. We will strive to present our work in the most clear and comprehensive manner possible.
> > > > > > >
> > > > > > > Once again, thank you for your valuable feedback.

---

### Official Review · Reviewer_KLe1 · 2024-07-15

**Soundness:** 3
**Presentation:** 3
**Contribution:** 2
**Rating:** 5
**Confidence:** 5

**Summary:**

The paper introduces a novel event-based graph spatiotemporal sensitive transformer framework aimed at enhancing the efficiency of object detection in dynamic vision systems. This framework leverages the unique properties of event camera data by modelling event data through a graph structure and incorporates key components such as the spatiotemporal sensitivity module and temporal activation controller to improve spatial and temporal processing efficiency. Additionally, the integration of a lightweight, multi-scale linear vision transformer further enhances processing efficiency.

**Strengths:**

1. The introduction of the event-based graph spatiotemporal sensitive transformer framework, combining event data processing and graph neural network technology, brings new perspectives and methods to the field of object detection.

2. By leveraging the unique properties of event camera data and incorporating key components such as the spatiotemporal sensitivity module, efficiency in spatial and temporal processing is significantly enhanced, leading to more effective object detection in dynamic vision systems.

3. The integration of a lightweight, multi-scale linear vision transformer further boosts processing efficiency, enabling the framework to excel in handling large-scale continuous spatiotemporal data.

**Weaknesses:**

1.	Could the manuscript clarify the advantages of your model over the AEC+YOLOv5 method mentioned in reference [44], and how the combination of GNN + LinearViT demonstrates its effectiveness?

2.	It would be better to provide a concise visualization of the data to analyze whether the SSM and TAC modules genuinely simulate the natural prioritization of fast-moving objects within the field of view, while lowering the priority of slower objects, rather than merely presenting a seemingly reasonable narrative.

3.	Please provide a concise analysis of the description concerning the comprehensive utilization of temporal information mentioned in the contributions. Moreover, are SSM and TAC critical steps that enhance the effective and precise focus on targets within the event data? Additionally, please clarify whether the Connected Subgraphs Construction is referenced for conducting ablation experiments to demonstrate its value.

**Questions:**

1.	It would be better for the manuscript to provide more detailed experimental validation and comparative analysis data to support the effectiveness and superiority of the framework, such as the Nighttime Driving Detection dataset.

2.	According to the content of reference [44] in the paper regarding AEC+DETR, there are factual errors. Please verify the specific metrics of this method on the two datasets and make the necessary corrections in the corresponding text.

3.	The absence of visualized detection results on the dataset needs to be supplemented and enhanced.

**Limitations:**

See Weaknesses

---

> ### Author Rebuttal · Authors · 2024-08-07
>
> Thank you for your insightful feedback on our paper. We have revised the manuscript based on your suggestions and hope the changes meet your expectations. Please feel free to contact us with any questions or further information you might need during the review process.
>
> **W1.** Regarding your suggestion to compare our method with the AEC+YOLOv5[2] approach, we find it highly valuable and will include this comparison in Table 1. Due to page constraints, we have included additional experimental data in the author rebuttal for your reference. To ensure a fair evaluation, we have replaced the detection head from RT-DETR[4] to the YOLO series, and specifically, following another reviewer's advice, to YOLOX[5], which is more commonly utilized in event camera-based object detection. On the Gen1 dataset, although our EGSST-B-Y method exhibits slightly lower detection accuracy compared to the AEC+YOLOv5 method, it demonstrates a 5% improvement in inference speed and is notably more lightweight. Similarly, on the 1Mpx dataset, the results are consistent, with our method achieving more than double the inference speed of the AEC approach.
>
> In the field of event data processing, we employ the Graph Transformer to effectively capture the spatiotemporal characteristics of event data using graph structures, while leveraging the Transformer's advantages in multi-scale information fusion and global feature extraction. Moreover, the linear nature of our model ensures high processing efficiency. As indicated by the comparisons in Table 1, although our model does not achieve the best performance on all evaluation metrics, its remarkably low inference time and parameter count demonstrate its effectiveness and significant contribution to the field of event data processing.
>
> **W2.** Thank you very much for the reminder that we have added an image of the visualisation results of SSM in the rebuttal pdf.
>
> **W3.** Thank you for your insightful questions and here are the relevant responses:
>
> 1. **Utilization of Temporal Information in GNNs:** Our research utilizes Graph Neural Networks (GNNs) to effectively process spatiotemporal information, preserving the temporal attributes of event data throughout the transformation process. We employ an event-level transformation strategy for converting graph features to frame features, which significantly retains temporal information and boosts adaptability to dynamic environments.
>
> 2. **Roles of the SSM and TAC Modules:** Our ablation studies have confirmed the pivotal roles of the SSM and TAC modules in focusing accurately on targets within event data. Removing the TAC module notably decreases precision, highlighting its essential role in enhancing detection accuracy.
>
> 3. **Role of Connected Subgraphs:** The construction of connected subgraphs is crucial in filtering out noise and sharpening the model’s focus. Experiments show that processing 10,000 event points retains approximately 73% of events, which helps diminish noise interference. Eliminating smaller subgraphs allows the model to concentrate on those with critical labels, thereby boosting detection efficiency and accuracy. Visualizations in the SSM+TAC section demonstrate the distribution of these subgraphs effectively.
>
> We appreciate your attention to these aspects of our work and hope these explanations address your queries satisfactorily.
>
> **Q1.** Thank you for your suggestions. Regarding the nighttime driving detection dataset, we find that the MVSEC-NIGHTL21 dataset [1,2] is an excellent choice. However, we regret not conducting experiments with this dataset due to time constraints and our analysis. Our method, along with the AEC[2] and ITS[3] methods, are event-based and do not rely on RGB frame information. Effective detection can be performed as long as there is sufficient event data. Furthermore, research on the paper [2] shows that while AEC and ITS experienced a decrease in detection accuracy on the MVSEC-NIGHTL21 dataset, they still maintained a high level of performance, demonstrating that event-based detectors are almost independent of RGB information.
>
> **Q2.** We are very sorry for this writing error and will make corrections in the table and the corresponding text in the paper.
>
> **Q3.** Thank you very much for your kind reminder. We have added the visualization results of our experiments on the relevant dataset. The results can be viewed in the rebuttal PDF.
>
> **References**:
>
> [1] The Multivehicle Stereo Event Camera Dataset: An Event Camera Dataset for 3D Perception, 2018
>
> [2] Better and Faster: Adaptive Event Conversion for Event-Based Object Detection, AAAI 2023
>
> [3] Inceptive event time-surfaces for object classification using neuromorphic cameras, ICIAR 2019
>
> [4] Detrs beat yolos on real-time object detection, CVPR 2024.
>
> [5] Yolox: Exceeding yolo series in 2021, arXiv 2021.

---

> > ### Comment · Reviewer_KLe1 · 2024-08-13
> >
> > Thank you for the rebuttal. It would be better for the author to provide their results on the Nighttime Driving Detection dataset in their final version. I tend to vote for a weak accept.

---

> > > ### Author Response · Authors · 2024-08-13
> > >
> > > Thank you very much for your suggestion. We have been actively conducting additional tests on relevant Nighttime Driving Detection datasets as per your request.

---

> > > > ### Author Response · Authors · 2024-08-14
> > > > **Sincere Appreciation and Humble Reminder**
> > > >
> > > > Dear Reviewer,
> > > > Thank you for taking the time to review our work! Your valuable and positive feedback is greatly appreciated. We noticed that the score has not been updated. It seems there is no final rating box this time, so the score may need to be adjusted in the original rating box. We would greatly appreciate it if you could make the adjustment.
> > > > Best regards.
> > > > The Authors

---

> ### Comment · Area_Chair_AW4a · 2024-08-12
>
> Dear Reviewer
>
> The author-discussion period ends at August 13, which is just 1 days away. Can you please discuss the rebuttal and the paper with the authors? Was there any concern that the authors' rebuttal did not address? Do you need further clarification from the authors?
>
> Best Regards,
>
> Your AC

---

### Author Rebuttal · Authors · 2024-08-07

We thank all reviewers for their very informative feedback. We have responded separately to each reviewer and attached a PDF file with figures and tables to enhance our rebuttal. Additionally, considering the page limitations of the PDF, we have briefly presented the new experimental data from Table 1 here for clarity. More detailed modifications will be reflected in the final manuscript. We appreciate all feedback from the reviewers and will incorporate it into the final version of our manuscript.

The additional experiments in Table 1 are listed below:

| Methods                | Dataset     | mAP (%)     | Time (ms) | Params (M) |
| ---------------------- | ----------- | ----------- | --------- | ---------- |
| EGSST-B                | Gen1 / 1Mpx | 44.6 / 45.4 | 4.6 / 5.1 | 3.5        |
| EGSST-E                | Gen1 / 1Mpx | 49.6 / 50.2 | 6.0 / 6.3 | 12.3       |
| EGSST-B-Y (with YOLOX) | Gen1 / 1Mpx | 43.9 / 44.1 | 3.7 / 5.0 | 2.9        |
| EGSST-E-Y (with YOLOX) | Gen1 / 1Mpx | 47.8 / 48.4 | 4.2 / 5.3 | 10.4       |

---

### Decision · Program_Chairs · 2024-09-25

**Decision:**

Accept (poster)

**Comment:**

This work proposes a novel event-based graph spatiotemporal transformer for object detection. It achieves SOTA performance with low inference time.

The strengths of this work are listed as follows.

1. Graph Neural networks for event-based object detection are novel and promising.
2. The spatiotemporal sensitive module (SSM) mimics the behavior of the human eye, which is suitable for event-based data.
3. The proposed method achieves SOTA in event-based object-detection tasks with low inference time.

The weaknesses of this work are listed as follows.

1. Dynamic Label Augmentation seems to be an important part of the proposed approach, but the authors do not provide its details. The details and ablation study about it should be conducted. The authors mentioned it in the rebuttal; please provide more details in the new version of this work.

2. In this work, an aggregation function is used to alleviate the burden of data processing. An ablation study should be conducted to evaluate its impact. It is unclear where the proposed method improves time efficiency. The authors refering the computation cost to another paper, it would be better to include such study in this work.

3. This work claims to be the first work to apply GNN to event-based object detection. However, it merely links feature maps generated by GNN to DETR. The authors stated in the rebuttal that they would revise such a claim.

Please take into account the reviews for the camera-ready version. Especially giving more details about each module/function used in this work, conducting necessary experiments to evaluate the impact of the aggregation function, soften the claims, etc.